# The Effect of the Comprehensive Reform of Agricultural Water Prices on Farmers’ Planting Structure in the Oasis–Desert Transition Zone—A Case Study of the Heihe River Basin

**DOI:** 10.3390/ijerph20064915

**Published:** 2023-03-10

**Authors:** Guifang Li, Dongdong Ma, Cuiping Zhao, Hang Li

**Affiliations:** 1Institute of Economics and Management, Henan Agricultural University, Zhengzhou 450046, China; 2School of Economics, Henan University of Economics and Law, Zhengzhou 450046, China; 3School of Economics and Management, University of Science and Technology Beijing, Beijing 100083, China

**Keywords:** tiered water price policy, uniform water price, agricultural water pricing method, farmers’ planting structure, Heihe River Basin

## Abstract

The comprehensive reform of agricultural water prices is an important policy for promoting the high-quality sustainable development of agriculture and ensuring national water security. In this study, based on farmer survey data from different water price policy implementation areas in the oasis–desert transition zone of the Heihe River Basin (HRB), crops are divided into high-water-consuming crops and low-water-consuming crops based on the average water consumption per hm^2^. The content of this study consists of two main parts: first, the study explores the response of farmers to different agricultural water price policies by comparing the impact of uniform water price and tiered water price policies on their planting structure. Second, it studies the areas where the tiered water price policy is implemented to verify the impact of price signals on farmers’ production decisions. The results show that, compared with the uniform water price policy, the implementation of the tiered water price policy will significantly reduce the proportion of high-water-consuming crops planted when other conditions remain unchanged. Under the tiered water price policy, the increase in water prices will reduce the proportion of farmers planting high-water-consuming crops, but the difference is not significant. This result reveals that when the opportunity cost of irrigation water increases, farmers will increase the proportion of low-water-consuming crops. The findings also indicate that a higher educational level, improved land inflow, the number of crop types, and satisfaction with the current subsidy policy will help increase the proportion of low-water-consuming crops. However, an increase in the family-cultivated land area will reduce the area of low-water-consuming crops.

## 1. Introduction

Not only are China’s water resources in short supply, but the water conflict between sectors is also relatively prominent. The per capita water resources are only one-quarter of the world’s per capita level, and the average annual water shortfall is 53.6 billion m^3^. Nearly two-thirds of Chinese cities are in a state of water shortage to varying degrees [1,2]. Agriculture is the largest water user in China, but water use efficiency in this sector has not been high. In 2020, the total national water consumption was 581.3 billion m^3^, and agricultural water consumption accounted for 62.14%, although agriculture contributed only 7.7% of GDP. Improving the agricultural water price policy, improving the efficiency of agricultural water use, and reducing the total amount of agricultural water use are necessary to alleviate the shortage of water resources in China [3,4].

The comprehensive reform of agricultural water prices in China has roughly gone through two main stages: complete free access to water supply prices has gradually been incorporated into the national commodity price management system [5]. There is a long-term imperative to promote high-quality and sustainable agricultural development and ensure national water security [6]. In 2004, the General Office of the State Council issued the “Notice on Promoting Water Price Reform to Promote Water Conservation and Protection of Water Resources”, which for the first time determined that the strategic goal of China’s water price reform is to allow full play to the role of the market mechanism and optimize the allocation of water resources. In 2012, the State Council issued the “Opinions on the Strictest Water Resources Management System”, which set “three red lines (total water consumption control, water use efficiency control, and water function zones)” and pointed out that it is necessary to adjust the collection standards and scope of water resource fees and strictly control the collection and use of water resource fees. In 2016, the “Opinions of the General Office of the State Council on Promoting the Comprehensive Reform of Agricultural Water Price” pointed out that it would take 10 years to establish and improve agricultural products that truly reflect the cost of water supply and to promote water conservation and the construction of farmland water conservancy facilities. The issuance and implementation of this document are of immeasurable significance in guiding the reform of agricultural water prices and improving the utilization rate of agricultural water resources with the operating mechanism of the market economy. In 2018, the National Development and Reform Commission issued the “Notice on Intensifying the Promotion of the Comprehensive Reform of Agricultural Water Prices”, which regards the comprehensive reform of agricultural water prices as the “bull nose” of agricultural water-saving work. The implementation of a water rights trading price is the goal of the comprehensive reform of agricultural water prices. The specific measures are as follows: on the basis of water rights distribution, farmers can obtain water rights quotas at low prices or free of charge, and when their water consumption is lower than their water rights quotas, they can obtain subsidies or rewards or sell the saved water. When their water consumption exceeds their quota of water rights, the price must be increased or water must be purchased at the market price. The central government has always emphasized the importance of increasing farmers’ income and ensuring food security. Therefore, it is generally believed that in addition to achieving the goal of water saving, the comprehensive reform of agricultural water prices should ensure that farmers’ income and agricultural product supply will not be greatly reduced in order to maintain social stability and national food security [7]. In the long run, this reform can be achieved through research and development and promotion of water-saving technologies (variety, equipment, infrastructure), but in the short term, a better approach is to adjust the crop planting structure to reduce agricultural irrigation water consumption. Under different water price policies, there are differences among crops in the cost of irrigation water, and changes in the planting area vary, thus affecting the planting structure [8].

Scholars have conducted many studies on the impact of water price policy on agricultural production in China and abroad. Early research focused mainly on measuring the price elasticity of demand for irrigation water to quantify the impact of water price policies on the development of irrigated agriculture and the choice of irrigation technology [9,10]. However, there is no uniform method or consensus on how to measure the price elasticity of the demand for irrigation water. Some scholars believe that when the price of water is low [11,12] or when water is rationed [13], water demand is inelastic, and changes in water prices affect irrigation water. The distribution effect is also small. Some studies have determined that the price elasticity of demand for water is negative; that is, an increase in water prices will lead farmers to reduce irrigation water consumption, indicating that the adjustment of water price policy can achieve the purpose of saving water to a certain extent [14]. Studies on the impact of water price on the crop planting structure have continued to appear. Some scholars have proposed that the water demand of farmers can be divided into the water demand of different crops, and the water demand of crops can be divided into land allocation demand and short-term water demand, while responses to changes in water prices are determined largely by the reallocation of land among crops [12,15]. Some researchers have concluded that the impact of water prices on farmers’ crop selection is not significant, and the fluctuation of crop rotation or agricultural product prices is more significant than the impact of water prices [16]. Others believe that changes in water prices will affect farmers’ crop choices and will produce substitution effects among crops [15,17,18]. The literature has often illustrated the impact of water price changes on the crop planting structure by verifying the impact of water prices on a single crop planting choice while failing to perform a comparative analysis between different crops and overlooking the marginal benefit and marginal cost analysis of farmers’ water use. This study attempts to compensate for the insufficiency of previous research, studying farmers’ planting choice behavior in different water price policy areas, paying attention to differences in the impact of different water price policies on farmers’ planting structure, and exploring the impact mechanism of water price changes.

“Oasis with water, desert without water”: the oasis–desert transition zone is the most active zone in the process of oasis creation and desertification. Inland arid areas in Northwest China account for 33% of the country’s land area, but the total water resources are less than 5% of the country’s total. Oasis agriculture accounts for only 5% of the total area nationwide, but more than 90% of the population is gathered in this area, contributing 83% of the GDP and making it the main space in which people survive and develop [19]. The average proportion of agricultural water consumption over the years has been as high as 90%, which has seriously crowded out other sectors’ water use. Changing the crop planting structure, improving the efficiency of irrigation technology, and reducing agricultural water use are inevitable ways to alleviate the regional water crisis [20]. The Heihe River Basin (HRB) is the second-largest inland watershed in Northwest China. It has a long history of oasis agriculture and is an important commodity grain base in the country. To alleviate the water conflict among the midstream departments, at the beginning of the 21st century, the midstream city of Zhangye started the construction of China’s first water-saving society. Market and other measures have largely optimized the planting structure and improved water use efficiency but have failed to reduce agricultural water demand [21]. Since the implementation of the “strictest water resources management system” in 2012, the total amount of water used in the middle reaches of the HRB has been continuously reduced, but the demand for agricultural water has continued to grow. From 2000 to 2018, the total sown area of crops in Zhangye city in the middle reaches of the HRB increased from 183,620 hm^2^ to 328,334 hm^2^, an increase of nearly 80%, and the phenomenon of the “water rebound” of agricultural water use did not decrease but increased.

As one of the first pilot areas for the comprehensive reform of agricultural water prices in the country, the HRB has implemented a water price policy of “total control and quota management”. Although the HRB has formulated detailed rules for the implementation of water rights transactions and clarified the subject and scope of such transactions, because the water rights market transaction price mechanism is still being explored, there are still many obstacles to water rights transactions between farmers. In addition, good irrigation water-metering facilities and end-use water management are helpful for the implementation of agricultural water pricing policies. The water conservancy management department of the HRB has installed facilities for measuring water volume at the mouth of the branch canal and has set up water pipe stations at the grassroots level. The water pipe stations are specifically responsible for the management of agricultural irrigation water, including transporting and monitoring irrigation water, communicating the current year’s water fee collection standards, accounting for farmers’ water consumption, collecting water fees, and overhauling irrigation facilities. Accurate measurements of agricultural irrigation water also contribute to the implementation of metering and charging policies. In addition, due to the drought and low rainfall in the oasis–desert transition zone of the HRB, evaporation is substantial, and the planting industry basically needs to rely on irrigation to develop. Therefore, farmers in the oasis–desert transition zone of the HRB are an appropriate research object, and the research results have reference significance for the implementation of water price reform in water-deficient areas dominated by agriculture.

Due to the importance of securing farmers’ income and the supply of agricultural products, it is impossible to require that agricultural water use meet the same economic efficiency standards as industrial and domestic water use. Under different water price policies, the marginal cost of water use by farmers is different. If a tiered water price policy is implemented, it is necessary to ensure that farmers can use a certain amount of low-priced water, which is equivalent to a subsidy and may lead them to make different decisions. Even under the same water price policy, farmers may respond differently due to their heterogeneity and differences in the water use efficiency of different crops [22]. In this study, based on farmer survey data of different water price policy implementation areas in the oasis–desert transition zone of the HRB, crops are divided into high-water-consuming crops and low-water-consuming crops according to average water consumption per hm^2^. The content of this study consists of two main parts: first, exploring the response of farmers to different agricultural water price policies by comparing the impact of the uniform water price and tiered water price policies on the planting structure and second, studying the areas where the tiered water price policy is implemented to verify the impact of price signals on farmers’ production decisions.

As the water shortage crisis becomes increasingly serious, the comprehensive reform of agricultural water prices is being implemented nationwide, which means that the price of agricultural irrigation water will be further increased. Therefore, the aim of this study is to optimize and improve the comprehensive reform of agricultural water prices for water resource management departments, especially for arid and semi-arid regions where irrigation agriculture is the mainstay. We discuss the mechanism of the impact of different water price policies and agricultural water pricing methods on the crop planting structure. The remainder of this study is organized as follows. Section 2 introduces the study area, analytical framework, and measurement model. Section 3 describes the impact of agricultural water price policy on farmers’ planting structure. Section 4 and Section 5 provide the discussion and conclusions.

## 2. Study Area and Methodology

### 2.1. Study Area and Farmer Survey

The HRB is located in the middle part of the Hexi Corridor in the arid region of Northwest China. Geographically, there are three major geomorphological units in the region. From south to north, they are the southern Qilian Mountains in the upstream region, the middle Hexi Corridor, and the northern Alxa High Plain in the lower reaches [23]. The HRB covers an area of approximately 130,000 km^2^. The midstream and part of the upstream region of the HRB are located in Zhangye city, Gansu Province, and the downstream region is located in the Ejin Banner, Inner Mongolia. Zhangye city is an irrigated agriculture economic zone that consists of six counties/districts: Ganzhou and Linze in the plain irrigation zone (PIZ), Minle and Shandan in the mountain irrigation zone (MIZ), and Gaotai in the northern desert irrigation zone (NDIZ). Due to differences in natural conditions, the crop planting structure is obviously different among irrigation zones. Seed maize, maize, and vegetables are mainly grown in the PIZ, cotton, maize-wheat intercrop, and seed watermelon are mainly grown in the NDIZ, and wheat, potato, maize, and barley are mainly grown in the MIZ. Linze in the PIZ and Gaotai in the NDIZ are located at the edge of the oasis in the midstream of the HRB. Linze County mainly grows seed maize, maize, and vegetables [13]. The cropland area in 2018 was 30,180 hm^2^, an increase of 72.45% over the area in 2000. Gaotai County is the main planting area for maize, seed watermelon, cumin, and tomato. The cropland area here in 2018 was 38,327 hm^2^, an increase of 77.27% over the area in 2001. These two counties/districts are the main regions experiencing deterioration of the ecological environment due to the expansion of an oasis. The shadow price of agricultural water in the different irrigation zones is approximately 0.22 RMB per m^3^, which is basically equal to the agricultural water price [24]. This finding indicates that there is a surplus of water resources compared to land resources.

Since 2017, Linze has implemented a comprehensive reform of agricultural water prices. That is, based on an agricultural water price of 0.168 RMB per m^3^, if the actual irrigation water used for seed maize or maize exceeds 10–30% of the irrigation quota, the charge for the excess irrigation water will be 1.2 times the water price. If it exceeds 30–50% of the irrigation quota, the charge for the excess irrigation water will be 1.5 times the water price. If it exceeds 50% of the irrigation quota, the charge will be 2 times the water price. According to the “Industrial Water Quota of Gansu Province”, the irrigation quota for corn is 7200 m^3^ ha^−1^. Therefore, if the amount of crop irrigation water does not exceed 7200 m^3^ ha^−1^, the irrigation cost will not be higher than 1383.6 RMB ha^−1^. If it does not exceed 10–30% beyond the irrigation quota, the irrigation cost will be 1383.6–1862.25 RMB ha^−1^. If it does not exceed the irrigation quota by 30–50%, the irrigation cost will be in the range of 1862.25–2253.9 RMB ha^−1^. If it exceeds the irrigation quota by 50%, the irrigation cost will exceed 2253.9 RMB. In fact, the irrigation costs for most farmers are more than 2253.9 RMB ha^−1^. Different from Linze County, Gaotai implements a uniform water price, and the water price is 0.218 RMB per m^3^. In summary, the sample sites involve the selection of farmers in the towns of Pingchuan, Liaoquan, and Yanuan, Linze County, and farmers in the town of Luocheng, Gaotai County [25].

To collect data for the current study, a survey was conducted in July and August 2019, and a well-designed and pretested questionnaire was administered to a sample taken from the population of the typical irrigation zones at the edge of the oasis. The stratified random sampling method was used in the farmer survey. Stratification ensures that the selected survey sample points can comprehensively cover the study area and are representative to reflect the impact of different water price policies on the study area. Random sampling involves randomly selecting sample villages and survey farmers after determining the survey township/town. It ensures the independence and representativeness of the samples to better reflect the overall characteristics. The questionnaire includes three main parts. The first is the basic information of the family (the age of the respondents, educational level, family population, family income and expenditure, subsidies, etc.). The second is the farmer’s planting structure and irrigation water use, mainly to conduct detailed investigations on the farmers’ irrigation water fee, crop planting structure, crop planting scale, and irrigation method. The third is the situation of household consumption.

After screening and removing incomplete questionnaires, 679 questionnaires were obtained from 22 villages, 4 towns, and 2 counties. The effective response rate was over 94%. There were 312 questionnaires from the town of Luocheng, Gaotai County, and 367 questionnaires from the towns of Pingchuan, Liaoquan, and Yanuan, Linze County. Due to the complexity of the questionnaire and the inclusion of rural and less educated people, face-to-face interviews were conducted. By providing clarification, the interviewers could assist the respondents and thus reduce any problems related to understanding the questions. The survey was carried out by well-trained university graduates. The survey site and sample distribution are shown in Figure 1.

Based on the survey data, the water consumption per hm^2^ of different crops was calculated. The water consumption per mu of crops was calculated based on the irrigation fee per mu and the farmer’s water fee collection method. Specifically, in areas where a uniform water price policy was implemented, the water consumption per hm^2^ of crops was calculated by dividing the irrigation cost per mu of crops by the water price per unit. In areas where step-type charging was implemented, there were two situations: first, for farmers whose water consumption per hm^2^ exceeded the low-price water quota, the irrigation fee exceeding the low-price water quota was divided by the price of high-price water and then added to the low-price water quota, which is the water consumption per hm^2^ of crops. Second, for farmers whose water consumption per mu did not exceed the low-price water quota, the irrigation cost per hm^2^ was divided by the low-price water price, which is the water consumption per mu of crops. The crop structure of the study area and the calculated water consumption per hm^2^ of crops are shown in Table 1.

Table 1 shows that maize, seed maize, vegetables, oil crops, and other seed crops were the main crops in the study area. Among them, the water consumption of maize and seed maize were 14,610 m^3^ per hm^2^ and 14,475 m^3^ per hm^2^, respectively. However, seed maize is one of the main economic crops supporting the development of the regional agricultural economy, and its net income per unit of water is higher than that of maize. Therefore, in the subsequent empirical analysis, maize was taken as the standard, and maize and crops with higher water consumption per hm^2^ were defined as high-water-consuming crops. In contrast, crops with lower water consumption per hm^2^ than maize were defined as low-water-consuming crops. Farmers’ planting choice behavior was analyzed on this basis.

### 2.2. An Analytical Framework for the Impact of Water Prices on Farmers’ Planting Structure

#### 2.2.1. Irrigation Water Cost of Farmers under Different Water Price Policies

The comprehensive reform of agricultural water prices mainly includes three kinds of policies, namely, the uniform water price, tiered water prices, and water rights transaction prices. Under the uniform water price system, the agricultural water fee is charged according to the uniform price. In contrast, the tiered water prices involve the implementation of classified metering charges and over-quota progressive price increases for agricultural irrigation water. A low water price and water quota are set per hm^2^ of land. Irrigation water within the quota is charged at the low water price, and the portion exceeding the quota is charged at the high water price. Unlike the other two types of water prices, the water rights transaction price is based on the initial allocation of agricultural water rights. Irrigation water consumption is subject to a low water price within the quota of the water rights certificate, and a high water price will be charged for the part exceeding the quota of the water rights certificate; farmers can sell the surplus water rights over the water rights certificate quota. In contrast to areas using the tiered water price, farmers in areas where the actual water rights transaction water price is implemented can obtain income by selling the saved water rights [2].

If *P_w_* is used to represent the water price under the uniform water price policy, *P_w_*_0_ is the water price within the low-priced water quota under the implementation of the tiered water price policy and the water price within the quota of the water warrant under the implementation of the water rights transaction policy, *P_w_*_1_ represents the price for water that exceeds the quota of low-priced water under the tiered water price policy and water that exceeds the quota of the water rights certificate under the water rights transaction price policy, *w*_0_ represents the low-priced water consumption quota per hm^2^ under the implementation of the tiered water price policy and the water consumption quota per hm^2^ of the water rights certificate under the water rights transaction water price policy, *w* represents the actual agricultural irrigation water consumption per hm^2^ of farmers, and *R* represents the income from the sale of surplus water rights by farmers in the water rights reform area. Then, in areas where the uniform water price policy is implemented, the irrigation water cost *C_w_* per hm^2^ of farmers can be expressed as:(1)$$C_{w}=P_{w}\times w$$
where *P_w_* is the marginal cost of irrigation water for farmers under the uniform water price.

In areas where tiered water prices are implemented, when the actual irrigation water consumption per hm^2^ of farmers does not exceed the quota of low-priced water per hm^2^, that is, when *w* ≤ *w*_0_, the irrigation water cost *C_w_* per hm^2^ of farmers can be expressed as:(2)$$C_{w}=P_{w0}\times w$$

When the actual irrigation water consumption per hm^2^ of farmers exceeds the quota of low-priced water per hm^2^, that is, when *w* > *w*_0_, the irrigation water cost *C_w_* per hm^2^ of farmers can be expressed as:(3)$$C_{w}=P_{w0}\times w_{0}+P_{w1}\times \left(w-w_{0}\right)$$

In areas where the water rights transaction price is implemented, assuming that only water transactions between farmers and other water users are considered, the market transaction price of farmers selling water is consistent with the additional water price stipulated by the government beyond the water rights certificate, that is, *P_w_*_1_ is farmers’ profit from selling water.

When the actual irrigation water consumption per hm^2^ of a farmer does not exceed the water usage quota of the water rights certificate, that is, when *w* ≤ *w*_0_, the farmer’s net income from the sale of irrigation water per hm^2^ is:(4)$$R=\left(P_{w1}-P_{w0}\right)\times \left(w_{0}-w\right)$$

The farmer’s agricultural irrigation water cost per hm^2^ is:(5)$$C_{w}=P_{w0}\times w-R=P_{w0}\times w_{0}+P_{w1}\times \left(w-w_{0}\right)$$

When the actual irrigation water consumption per hm^2^ of the farmer exceeds the water quota of the water rights certificate, that is, when *w* > *w*_0_, the agricultural irrigation water cost per hm^2^ of the farmer is:(6)$$C_{w}=P_{w0}\times w_{0}+P_{w1}\times \left(w-w_{0}\right)$$

Regardless of whether farmers’ irrigation water consumption exceeds the water rights quota, the cost model for farmers under the water rights transaction price is the same. The cost models in Equations (5) and (6) can be transformed into the following form:(7)$$C_{w}=P_{w1}\times w-\left(P_{w1}-P_{w0}\right)\times w_{0}$$

Equation (7) shows that under the water rights transaction price, the marginal cost of irrigation water for farmers is the high water price *P_w_*_1_; $\left(P_{w1}-P_{w0}\right)\times w_{0}$ is the difference between the market sales revenue of the water rights quota and the cost of purchasing the water rights quota at low prices, which can be regarded as a fixed subsidy from the government, that is, the net income that farmers can obtain regardless of whether they plant, what crops they plant, and how much water they irrigate.

#### 2.2.2. Influence of Water Price Policy and Pricing Method on Farmers’ Planting Structure

① Impact of water price on farmers’ planting structure. If the water price is regarded as the marginal cost of irrigation water for farmers, then all other factors that affect farmers’ production decisions can be regarded simply as factors affecting the marginal benefit of irrigation water. Under certain conditions of technology, the short-term impact on farmers of an increase in the marginal cost of water may be reflected mainly in the adjustment of the planting structure, while the long-term impact includes the innovation and promotion of water-saving technologies. From the perspective of the marginal cost of water, the marginal cost under the uniform water price policy is the direct pricing of water, while the marginal cost under the water rights transaction water price policy is the high price of water, and the marginal cost under the tiered water price policy depends on whether the water consumption exceeds the low price. The marginal cost within the water use quota is the low water price, and the marginal cost exceeding the quota is the high water price. Low-priced water and water quotas are the basis for ensuring farmers’ income and the agricultural product supply, while allowing farmers to face higher marginal costs of water use is the “bull nose” that guides them to optimize the structure of agricultural production, adopt water-saving technologies, and improve water use efficiency.

When water prices rise, farmers may choose to plant low-water-consuming crops instead of high-water-consuming crops. This is because rising water prices will reduce production profits, the production profits of high-water-consuming crops will fall even more, and farmers will switch to growing crops that consume less water per hm^2^ and are relatively more profitable.

② Impact of the pricing method on farmers’ planting structure. According to Equation (7), the marginal cost of water consumption for farmers under the water rights transaction price remains unchanged, which is the same as the situation under the uniform water price; however, there is a fixed subsidy for farmers under the water rights transaction price. Even if the price of water under the uniform water price policy is equal to the high water price under the water rights transaction water price policy, but their impact on agricultural production may be different. The reason is that the fixed subsidy under the water price policy for water rights trading may have an additional impact on the crop planting structure by easing farmers’ liquidity constraints. This additional impact may take the form of enabling farmers to invest in changing varieties, thereby encouraging them to grow low-water-consuming crops, or it may take the form of enabling farmers to pay high water prices or invest in water-saving technologies, allowing them to maintain water-intensive crops.

The impact of tiered water prices on farmers’ crop planting structure is more complicated. If the quota of low-priced water per hm^2^ of cultivated land is so large that the water consumption of each crop will not exceed it (which does not occur in reality), then only the price level of low-priced water affects the crop structure. Conversely, if the quota is so low that the water consumption for each crop exceeds it, then, as in the case of trading water prices under the water rights system, only high water prices are valid price signals. The difference between the prices and the sizes of the quotas is related to the amount of the quota planting subsidy. If the low-priced water quota restricts only some crops, then the price difference between high-priced water and low-priced water and the size of the quota will have a direct impact on the crop planting structure. The underlying equilibrium mechanism is as follows: First, farmers will select the optimal crop types according to the price levels of low-priced water and high-priced water, corresponding to specific high-water-consuming crops and low-water-consuming crops, respectively. Second, farmers will combine the two types of crops to form a specific planting structure.

Equations (3), (5) and (6) show that the water price for water rights transactions can be regarded as a special ladder water price; the special feature is that the low-priced water quota is zero. According to the above conclusion that lowering the low-priced water quota will increase the proportion of low-water-consuming crops planted, if the water prices and quotas in the tiered water price and the water rights transaction water price are the same, then the water rights transaction water price policy will still be more effective than the tiered water price. It is beneficial for promoting the cultivation of low-water-consuming crops.

#### 2.2.3. Implementation of Water Price Policies in the Study Area and Research Hypotheses

According to the survey carried out by the research team in the oasis–desert transition zone of the HRB in 2019, although the water rights trading system had been implemented for more than 20 years, due to the imperfect water rights trading market, the transaction phenomenon among smallholder farmers had only been short-lived and had not formed a scale. At the time, the uniform water price and the tiered water price were the main types of water prices implemented in the oasis–desert transition zone of the HRB. The water price in areas where the uniform water price policy was implemented was between the low water price and the high water price implemented in the tiered water price system.

Theoretically, the low water price in the tiered water price system can reduce the total cost of water use for farmers, thereby ensuring the stability of their income and the supply of agricultural products, while the high water price can encourage farmers to save water and improve the planting structure, thereby improving the economic efficiency of irrigation water. According to the survey results, although a certain low-priced water quota would be set under the tiered water price policy, to encourage farmers to save water, the low-priced water quota per hm^2^ of arable land was often lower than the average water requirement per hm^2^ of most crops or even all common crops. Of course, it was possible that some farmers faced a higher quota of low-priced water or used less water due to the adoption of water-saving technologies and good soil and climatic conditions so that the water consumption of their crops was lower than the quota, but most farmers would use high-priced water. The high water price was the price signal that truly worked. Some farmers also reduced the weighted average water consumption per hm^2^ by planting crops with lower water consumption so that their water consumption per hm^2^ was equal to the quota.

From this, it can be speculated that, compared with areas where uniform water prices are implemented, high-priced water is the most important price signal in areas where tiered water prices are implemented. It is precisely because of the higher price level that policy goals can be better achieved. Accordingly, the following hypotheses are proposed:

**H1.** 
*The agricultural water price policy will affect farmers’ crop choices. Compared with areas where a uniform water price is implemented, farmers are more likely to plant low-water-consuming crops in areas with tiered water prices.*


**H2.** 
*In areas with tiered water prices, the higher the price of water is, the more likely farmers are to grow low-water-consuming crops.*


### 2.3. Measurement Model

According to the results of previous research [2,26] and the sample characteristics and considering that the salinity of groundwater in the oasis–desert transition zone of the HRB is relatively high, farmers mainly use flood irrigation, and there are few differences in irrigation water sources and irrigation techniques. In addition, the salinity of cultivated land in this area is relatively high, and there are also few differences in the impact of cultivated land quality on farmers. Therefore, in the selection of variables, the impact of factors such as irrigation water source, irrigation technology, and cultivated land quality was not considered. In this study, the household was used as the unit, the proportion of farmers’ high-water-consuming crop planting area to the household’s land management area was selected as the explained variable, and the farmers’ planting structure model was constructed.

#### 2.3.1. The Model of the Impact of Different Water Price Policies on Farmers’ Planting Structure

To compare the impact of different water price policies on farmers’ planting structure, the uniform water price policy and tiered water price policy were selected as key explanatory variables to analyze the impact of different water price policies on farmers’ planting choices.

The control variables were selected mainly from the characteristics of farmers’ population sociology, farmers’ production management characteristics, and farmers’ farming needs and attitudes toward risks. First, in terms of the sociological characteristics of the population of farmers, considering that farmers with rich agricultural production experience and a high educational level may choose to grow crops with higher water consumption but higher returns, age (lnAGE) and educational level (D3) were the two control variables [27]. At the same time, the production of different agricultural products requires different amounts of labor. Therefore, the variable of the labor quantity of family farming (lnAL) was added to the model [28]. Second, in terms of farmers’ production management characteristics, the larger the land management area of a family is, the more diverse the crop selection may be. Therefore, variables such as the family’s land management area (lnFLA) and land inflow (D4) were added to the model [29,30]. Third, in terms of farming needs and attitudes toward risks, it is generally believed that the degree of household dependence on agricultural income may affect farmers’ production decisions, so the variable of the proportion of family agricultural income (lnAIR) was selected [31]. At the same time, farmers are considered to avoid risks and pursue high returns, so indicators such as the number of crop species (lnCT) and satisfaction with the current subsidy policy (D5) were selected to measure the policy impact [25]. In addition, factors that affect farmers’ marginal income from irrigation water may affect their crop selection, but since the study area is a uniform market, the product price can be regarded as a constant; therefore, the product price variable was not included in the model. In summary, the model of the impact of different water price policies on farmers’ planting structure constructed in this paper is as follows:(8)$$Y_{i}=\beta_{0}+\beta_{1}D_{1}+\gamma X_{i}+\mu_{i}$$
where *Y_i_* represents the proportion of the farmer’s planting area of high-water-consuming crops to the household’s land management area; *D*_1_ represents different water price policies, the uniform water price, and tiered water prices; *X_i_* represents the main control variables, such as farmers’ age, educational level, the proportion of agricultural labor, the family-cultivated land area, land inflow, the proportion of agricultural income, the number of crop types and satisfaction with the current subsidy policy; and $\mu_{i}$ is the random disturbance term.

#### 2.3.2. Model of the Impact of the Tiered Water Pricing Method on Farmers’ Planting Structure

To further verify the impact of price signals on the planting structure and to exclude the impact of policy factors, farmers in areas with tiered water prices were selected as the research sample. Since the high price of water determines the marginal cost of irrigation water for farmers and the low-priced water quota per hm^2^ of cultivated land is a special form of subsidy, the quota is determined according to the “Industrial Water Quota in Gansu Province”, and the difference in the effect on farmers’ planting decisions is very small. However, in areas where the tiered water price is implemented, the high water price rises in steps, which affects farmers’ planting decisions by affecting their budget constraints. Therefore, the change in tiered water prices was selected as the key explanatory variable to analyze the impact of water price changes on farmers’ planting decisions. In summary, the model constructed in this study for the impact of water price on farmers’ planting structure is as follows:(9)$$Y_{i}=\alpha_{0}+\alpha_{1}D_{2}+\gamma X_{i}+\mu_{i}$$
where *D*_2_ represents the tiered water price, *D*_2_ = 0 is the water price below the quota; *D*_2_ = 1 is 1.2 times the water price; *D*_2_ = 2 is 1.5 times the water price; *D*_2_ = 3 is the between 1.5 times and 2 times the water price; and *D*_2_ = 4 is 2 times the water price.

#### 2.3.3. Descriptive Statistical Analysis of the Variables

The meaning and descriptive statistics of the variables are shown in Table 2 and Table 3. The analysis of the survey data shows that the proportion of high-water-consuming crop in farmers’ planting area is approximately 43%. In the areas where the tiered water price policy is implemented, the irrigation water used by farmers exceeds the irrigation quota (6900 m^3^ per hm^2^), and most farmers pay twice the water price.

The average age of farmers in the study area is approximately 53 years old, with the majority being between 45 and 65 years old. For most farmers, their educational level is junior high school or below. Regarding the production and operation of farmers, the average farmland area is approximately 1 hm^2^. Due to the influence of large planters, the maximum family farmland area can reach 15.33 hm^2^. The minimum is only 0.17 hm^2^. More than half of family farming income is agricultural income. Finally, the average number of crops grown by farmers is 2–3, and most farmers are satisfied with the current subsidy policy, which mainly includes agricultural subsidies, pension subsidies, grassland subsidies, and public welfare forest subsidies [32].

## 3. Results

The econometric software Stata 15.0 is used for analysis. After testing, the variance inflation factor between the influencing variables does not exceed 5, indicating that there is no multicollinearity. The model disturbance term obeys a normal distribution, and heteroskedasticity is processed by taking the logarithm and conducting a robustness test. Table 4 highlights the coefficients and significance of all the variables of the different models. From the regression results of each model, the significance and coefficient signs of the main variables are in line with expectations.

Model 1 and Model 2 are the regression results of the impact of the two water price policies and water pricing on farmers’ planting structure, respectively. The explained variables are the proportion of farmers planting high-water-consuming crops. The difference is that Model 1 has two key explanatory variables: different water price policies (*D*_1_), uniform water price and tiered water price, whereas the key explanatory variable of Model 2 is the tiered water price (*D*_2_). The purpose is to examine the impact of water price signals on the planting structure after excluding the influence of policy factors.

Model 1 shows that, compared with the implementation of the uniform water price policy, the implementation of the tiered water price policy has a significant negative impact on the planting structure. Specifically, the variable of whether the tiered water price policy is implemented is significant at the 1% level, and the coefficient is negative. This finding reveals that, compared with the uniform water price, the implementation of the tiered water price policy will reduce the proportion of high-water-consuming crops planted when other conditions remain unchanged. This result is in line with the expectations of this study, and H1 is therefore verified.

Model 2 explains that under the tiered water price policy, the increase in high water prices will reduce the proportion of farmers planting high-water-consuming crops, but the difference is not significant. The possible reasons are first, that the water price formulated under the current tiered water price policy cannot effectively encourage farmers to choose low-water-consuming crops; second, that due to the different water requirements during the growth of different crops, even if the water price is high, farmers will irrigate; and third, that the development of contract farming may affect the demand for agricultural irrigation water. In addition, the survey shows that the comprehensive reform of agricultural water prices will cause some farmers to give up planting. However, farmers who choose to continue farming will adjust their planting structure according to the policy and generally will not reduce their water demand due to rising water prices. This result is in line with the expectations of this paper; therefore, H2 is verified.

For the control variables, the findings indicate that first, educational level (*D*_3_) is significant at the 5% level in both models, and the sign of the coefficient is negative, indicating that the higher the educational level of farmers is, the better their understanding of the water price policy reform will be and the more low-water-consuming crops they will plant. Second, the sign of the indicator of family-cultivated land area (lnFLA) is positive in both models. The difference is that this indicator is not significant in Model 1, while it is significant at the 1% level in Model 2, which implies that an increase in family-cultivated land area will increase the planting area of high-water-consuming crops, and the operation of small farmers will help to improve the effect of policy implementation. Moreover, the sign of land inflow (*D*_4_) is opposite to that of family-cultivated land area; it is not significant in Model 1 but is significant at the 10% level in Model 2, indicating that farmers with strong planting willingness will reduce their area of high-water-consuming crops. Third, the number of crop types (lnCT) (significant at the 1% level in both models) and satisfaction with the current subsidy policy (*D*_5_) (significant at the 5% level in Model 1 and at the 10% level in Model 2) have negative signs in both models, indicating that the more varieties of crops farmers plant, the higher their satisfaction with the current subsidy policy will be, and the more likely they will be to increase the proportion of low-water-consuming crops, which is in line with the expectations of this study [25,27,28,29,30,31].

## 4. Discussion

### 4.1. Influence of the Water Rights Trading Price Policy on Farmers’ Planting Structure

The implementation of water rights trading water prices is the ultimate goal of China’s comprehensive reform of agricultural water prices. However, water rights transactions at the farmer level have not yet formed a scale. At the beginning of the 21st century, the HRB was committed to establishing a water rights trading market. However, due to the imperfect water pricing mechanism and the high cost of water rights trading, the effect of this market has not been obvious. Regarding relevant studies, for example, Dong et al. (2020) comparatively examined the impact of uniform water prices, tiered water prices, and water rights transaction water prices on farmers’ planting structure and pointed out that water rights transaction water prices can encourage farmers to choose more water-saving crops. However, the water rights transaction price mentioned in this article was a “water rights transaction price without a water selling mechanism”; the farmers in the surveyed areas had no water sales channels, and the sample villages had never conducted water rights transactions. Therefore, the water rights trading policy in this article was just another form of tiered water prices and in fact could not truly reflect the impact of the water rights trading policy on farmers’ planting structure. This study draws on the theoretical framework of Dong et al. (2020) and makes improvements on this basis to reflect the impact of the current water price policy and pricing method on the planting structure; therefore, it has more practical guiding significance [2].

### 4.2. Impact of the Comprehensive Reform of Agricultural Water Prices on Farmers’ Income

In 2016, the General Office of the State Council issued the “Opinions on Promoting the Comprehensive Reform of Agricultural Water Prices”, which marked the comprehensive reform of agricultural water prices being rolled out from the pilot program to the whole country. In recent years, the pilot areas have achieved remarkable results in exploring the comprehensive reform of agricultural water prices, and the sustainable utilization of agricultural water resources has ushered in new development opportunities. The comprehensive reform of agricultural water prices has made positive progress in improving the efficiency of water management, improving the water price formation mechanism, and promoting high efficiency and water conservation in agriculture. However, some scholars [26,32] have pointed out that an increase in irrigation water prices will reduce not only irrigation water use but also farmers’ planting income. Therefore, the state should appropriately subsidize irrigation water users to ensure stable grain production and ensure that farmers’ irrigation rights and interests are not infringed upon. The results of this study indicate that on the one hand, farmers in areas with tiered water prices will reduce the proportion of high-water-consuming crops they plant, and many low-water-consuming crops result in higher income per hm^2^. Therefore, to a certain extent, the tiered water price policy can increase planting income by optimizing the planting structure. On the other hand, with the increase in water prices, as the marginal cost of irrigation water for farmers continues to increase, farmers will abandon their farms and go out to work in other sectors, and their planting enthusiasm will decrease. Therefore, the comprehensive reform of agricultural water prices is a double-edged sword. In fact, it is necessary to grasp the range of agricultural water price increases and supplement it with compensation policies to save water without reducing farmers’ income.

### 4.3. Whether the Comprehensive Reform of Agricultural Water Prices has Achieved the Purpose of Saving Water

Using price leveraging to improve the efficiency of irrigation water use in the HRB is one of the main purposes of the comprehensive reform of agricultural water prices. However, raising water prices may not necessarily save water. Some scholars [33,34] believe that the water price policy cannot achieve a significant water-saving effect within a certain range because within a wide range of low prices, farmers’ water demand elasticity is very small. When the water price increases to a level that significantly reduces planting income, farmers will start to reduce their water consumption, and at the same time, their income level will decline. According to this study, on the one hand, farmers in areas where tiered water prices are implemented complained that the cost of irrigation water was too high. On the other hand, based on the water demand for crop irrigation, farmers irrigated as usual without reducing their demand for irrigation. Therefore, the current tiered water price did not achieve a good water-saving effect. If water is to be saved, it is necessary to further increase the water price. However, excessively high water prices will inevitably reduce the income of farmers and affect the social and economic development and food security of the region. Therefore, the water price formulation of tiered water prices is particularly important, not only to achieve water savings but also to take into account the sustainable development of the region, which is an important topic for future research.

## 5. Conclusions

Based on farmer survey data on different water price policy implementation areas in the oasis–desert transition zone of the HRB in 2019, this study constructs an analytical framework for the impact of water prices on farmers’ planting structure and builds an econometric model. The crucial findings are as follows. Compared with the uniform water price policy, the implementation of the tiered water price policy will significantly reduce the proportion of high-water-consuming crops planted when other conditions remain unchanged. Additionally, the pricing method of the tiered water price policy will reduce the proportion of farmers planting high-water-consuming crops. However, the difference is not significant, which reveals that when the opportunity cost of irrigation water increases, farmers will increase the proportion of low-water-consuming crops. Finally, the results suggest that the proportion of low-water-consuming crops can be increased by increasing farmers’ educational level, reducing the family-cultivated land area, improving land inflow, increasing the number of crop types, and increasing farmers’ satisfaction with the current subsidy policy.

## Figures and Tables

**Figure 1 ijerph-20-04915-f001:**
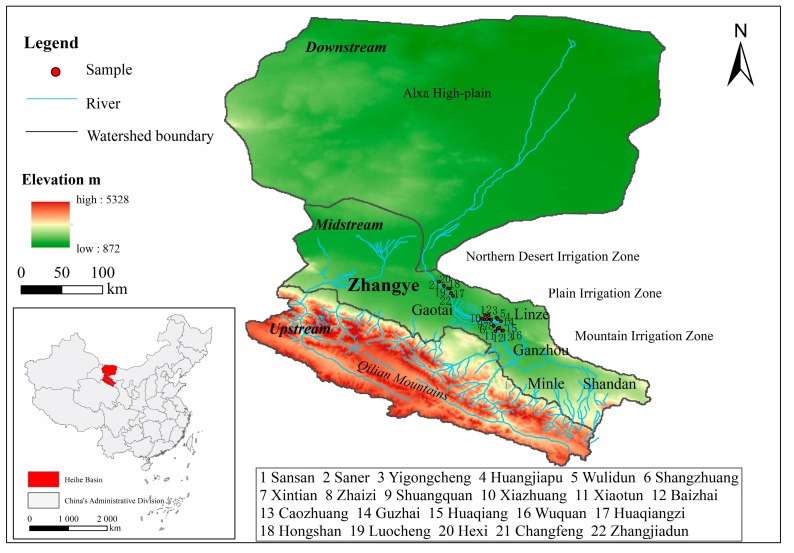
Survey site and sample distribution.

**Table 1 ijerph-20-04915-t001:** Crop structure and crop water consumption in the study area.

Crop Type	Crop Planing Area in 2018		Irrigation Water Per hm^2^ (m^3^/hm^2^)	Irrigation Cost Per hm^2^ (RMB/hm^2^)	The Net Income Per m³ (RMB/m³)
	Linze County (hm^2^)	Gaotai County (hm^2^)			
Maize	10,338	12,526	14,610	3615	1.32
Seed maize	12,996	3548	14,475	3570	2.38
Wheat	417	3246	10,020	2370	1.85
cumin			5610	1335	2.41
Tomato	Vegetables: 3604	Vegetables: 5829	9300	2205	4.18
Chili			14,715	3495	2.14
Seed watermelon		Other seed crops: 2962	6990	1800	8.05
Seed sunflower			6540	1545	4.65
Seed lettuce			11,625	2760	1.94
Seed gourd			11,685	2775	4.27

**Table 2 ijerph-20-04915-t002:** The main determinants of farmers’ planting structure and the meanings.

Variable Type	Variables	Coding	Description of the Variables	Effect Direction
Dependent variable	Proportion of high-water-consuming crops	*Y*	The dependent variable represents changes in the planting structure of farmers	
Key explanatory variables	Whether the tiered water price policy has been implemented	*D* _1_	*D*_1_ = 0 uniform water price policy; *D*_1_ = 1 tiered water price policy	Negative
	The pricing method of tiered water prices	*D* _2_	*D*_2_ represents the tiered water price; *D*_2_ = 0 is the water price below the quota; *D*_2_ = 1 is 1.2 times the water price; *D*_2_ = 2 is 1.5 times the water price; *D*_2_ = 3 is between 1.5 times and 2 times the water price; and *D*_2_ = 4 is 2 times the water price	Negative
Control variables	Age	lnAGE	This indicator mainly reflects the influence of farmers’ planting experience	Negative
	Educational level	*D* _3_	*D*_3_ = 1 means the farmer has received primary education or above; otherwise *D*_3_ = 0	Negative
	The proportion of agricultural labor	lnAL	This indicator mainly reflects the influence of the number of agricultural laborers on the farmers’ planting structure	Negative
	Family-cultivated land area	lnFLA	This indicator mainly reflects the impact of farmers’ production scale	Positive
	Land inflow	*D* _4_	*D*_4_ = 1 means farmers have land inflow; otherwise *D*_4_ = 0, which mainly reflects the impact of farmers’ farming needs	Negative
	The proportion of agricultural income	lnAIR	The logarithm of the proportion of household agricultural income, which reflects the influence of farmers’ dependence on agricultural income	Negative
	Number of crop types	lnCT	The logarithm of the number of types of crops planted by farmers; this indicator mainly reflects the influence of farmers’ attitudes toward risks	Negative
	Satisfaction with the current subsidy policy	*D* _5_	0 = Very dissatisfied; 1 = Dissatisfied; 2 = Indifferent; 3 = Satisfied; 4 = Very satisfied	Negative

**Table 3 ijerph-20-04915-t003:** Statistical descriptions of the main explanatory variables.

Variables	Mean	Max	Min	Standard Deviation
*Y*	0.4271	1.0000	0.0000	0.1927
*D* _1_	0.5397	1.0000	0.0000	0.4984
*D* _2_	3.9412	4.0000	2.0000	0.9894
*AGE*	52.7162	80.0000	19.0000	9.0697
*D* _3_	0.5132	1.0000	0.0000	0.4998
*AL*	0.5700	1.0000	0.0000	0.2165
*FLA*	16.2015	230.0000	2.5000	13.7929
*D* _4_	0.0662	1.0000	0.0000	0.2486
*AIR*	0.5401	1.0000	0.0000	0.2410
*CT*	2.5721	4.0000	1.0000	0.7845
*D* _5_	2.3859	4.0000	0.0000	0.7002

**Table 4 ijerph-20-04915-t004:** Coefficients and significance of the variables in the different models.

The Type of Explanatory Variables	Variables	Dependent Variable: Proportion of High-Water-Consuming Crops			
		Model 1		Model 2	
		Coefficient	Std.Err.	Coefficient	Std.Err.
	Constant	0.6789 (1.34)	0.5069	0.8610 (1.00)	0.8599
Key explanatory variables	*D* _1_	−0.8522 (−12.63 ***)	0.0675		
	*D* _2_			−0.0525 (−0.65)	0.0812
Control variables	*lnAGE*	−0.0606 (−0.52)	0.1161	−0.0264 (−0.17)	0.1560
	*D* _3_	−0.0823 (−2.01 **)	0.0410	−0.1370 (−2.39 **)	0.0574
	*lnAL*	−0.0046 (−0.09)	0.0494	0.0693 (0.71)	0.0977
	*lnFLA*	0.0063 (0.14)	0.0437	0.1665 (2.96 ***)	0.0562
	*D* _4_	−0.0566 (−0.68)	0.0830	−0.1889 (−1.81 *)	0.1046
	*lnAIR*	−0.0296 (−0.77)	0.0386	0.1327 (1.40)	0.0951
	*lnCT*	−0.8650 (−7.66 ***)	0.1129	−0.9612 (−4.76 ***)	0.2019
	*D* _5_	−0.0719 (−2.60 **)	0.0277	−0.0794 (−1.91 *)	0.0415
Log likelihood =		−445.8236		−249.0404	
LR chi2(9) =		214.33		42.92	
Prob > chi2 =		0.0000		0.0000	
Pseudo R2 =		0.1938		0.0793	
Number of obs =		679		367	

***, ** and * represent significance levels of 1%, 5%, and 10%, respectively; the t value is in parentheses.

## Data Availability

Data supporting the conclusions of this article are included within the article. The dataset generated and/or analyzed during the present study is available from the corresponding author.

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
