# Peer review of "The Effect of the Comprehensive Reform of Agricultural Water Prices on Farmers’ Planting Structure in the Oasis–Desert Transition Zone—A Case Study of the Heihe River Basin"

_ijerph, 2023, doi:10.3390/ijerph20064915_

Round 1
Reviewer 1 Report
The comments are as follows:
(1) In line 313, the survey was mentioned several times, but not much information was described at the moment. It is recommended to describe the survey in this manuscript or add a reference at least. Study area and farmer survey section should appear earlier.
(2) In line 334, the authors mentioned the “according to the research results and sample characteristics”, it is not clear what the results are.
(3) It is recommended to do some statistical analysis based on the survey data before proposing equation (8) and (9), I would suggest move section 3 before those equations.
(4) In line 447, it says that “the specific sample is shown in Figure 1”, however figure 1 is not what it should be. The questionnaire could be used as a supplementary material to help the reader better understand the whole process. It is not clear at all what kind of data has been collected at the moment. You should mention some details regards to the survey, like how many questionnaires were received.
(5) In lines from 494 to 498, there should be some results, like numbers, figures etc. In line 497 it says “table 4 highlights …”, there is no table 4 in this manuscript.
(6) In section 4, it is recommended to display the two models (equations) first.
(7) The conclusion part should focus more on the main finds of this manuscript.
Author Response
Responses to Reviewer 1’s Comments:
- In line 313, the survey was mentioned several times, but not much information was described at the moment. It is recommended to describe the survey in this manuscript or add a reference at least. Study area and farmer survey section should appear earlier.
Response: Thank you very much for your comment. Based on your suggestion, we have revised the content as follows: first, the second section is changed from "2 Materials and Methods" to "2. Study area and methodology" (line 187). Second, the contents of "3.1. Study area and farmer survey" and "3.2. Analysis of water consumption of crops" in the original manuscript were adjusted to "2.1. Study area and farmer survey" (lines 188-280). Third, the content in "3.3. Descriptive statistical analysis of variables" in the original manuscript was adjusted to "2.3.3. Descriptive statistical analysis of the variables" in the revised manuscript (lines 512-529). Fourth, the serial numbers of the titles of sections 3, 4 and 5 have been modified (lines 530, 589, 590, 610, 635 and 655).
- In line 334, the authors mentioned the “according to the research results and sample characteristics”, it is not clear what the results are.
Response: Thank you for this comment. We were not sufficiently clear, and “research results” refers to “the results of previous research”. The authors have revised the expression and added references [2,26] on line 445.
- It is recommended to do some statistical analysis based on the survey data before proposing equation (8) and (9), I would suggest move section 3 before those equations.
Response: Based on the reviewer’s comment, we have changed the contents of "3.1. Study area and farmer survey" and "3.2. Analysis of water consumption of crops" in the original manuscript to "2.1. Study area and farmer survey" (lines 188-280). Thank you.
- In line 447, it says that “the specific sample is shown in Figure 1”, however figure 1 is not what it should be. The questionnaire could be used as a supplementary material to help the reader better understand the whole process. It is not clear at all what kind of data has been collected at the moment. You should mention some details regards to the survey, like how many questionnaires were received.
Response: Thank you very much for your comments. We have supplemented and explained the process and content of the farmer survey. This paper introduces the method of the farmer survey, the selection of survey samples, the content and quantity of the questionnaire, the survey team and so on (lines 230-253).
- In lines from 494 to 498, there should be some results, like numbers, figures etc. In line 497 it says “table 4 highlights …”, there is no table 4 in this manuscript.
Response: Thank you very much for your comment. The serial number in Table 4 in the original manuscript was incorrectly marked, and we have revised it on line 586.
- In section 4, it is recommended to display the two models (equations) first.
Response: Based on the reviewer’s comment, we have modified and supplemented the content of “2.3. Measurement model”. First, the control variables represented by Xi in formulas (8) and (9) are supplemented, which can make the meaning of the formulas clearer (lines 489-493). Second, the content of section 3 in the original manuscript was adjusted to section 2 in the revised manuscript. Such modifications may enable a more appropriate link between the econometric model and the regression results. Thank you.
(7) The conclusion part should focus more on the main finds of this manuscript.
Response: Thank you for this comment. We have revised the content of the conclusion (lines 656-669).
Reviewer 2 Report
This is an interesting study. However, the author's research method and variable selection still have some problems. A few comments for your reference:
(1) The choice of research methods. The author seems to be conducting a policy assessment based on a cross-sectional data, which seems inadvisable in the current research context. The reasons are as follows: the adjustment of the impact of the policy on water price is instantaneous, which is a static time node data, while the adjustment of farmers' crop structure can be made in the current period or across periods. Therefore, the author's current research ideas and methods cannot accurately estimate the impact of water price adjustment on farmers' crop structure adjustment. To assess this impact, dynamic panel data should be used.
(2) The basis for variable selection needs further clarification. For example, the selection of control variables should refer to some relevant literature.
Author Response
- The choice of research methods. The author seems to be conducting a policy assessment based on a cross-sectional data, which seems inadvisable in the current research context. The reasons are as follows: the adjustment of the impact of the policy on water price is instantaneous, which is a static time node data, while the adjustment of farmers' crop structure can be made in the current period or across periods. Therefore, the author's current research ideas and methods cannot accurately estimate the impact of water price adjustment on farmers' crop structure adjustment. To assess this impact, dynamic panel data should be used.
Response: Thank you very much for your comments and professional review. These valuable suggestions served as inspiration for us. We fully agree with your viewpoint that "the adjustment of the impact of the policy on water price is instantaneous, which is a static time node data, while the adjustment of farmers' crop structure can be made in the current period or across periods". This is also one of the topics that we will study in the future. However, regarding the research content and method of this paper, we need to explain as follows:
First, rather than studying only one type of water price policy, the purpose of this paper is to study the impact of different agricultural water price policies (uniform water price policy and tiered water price policy) on farmers' planting structure in a certain research area to reveal which water price policy is more conducive to encouraging farmers to choose low-water-consuming crops. As mentioned in the article, under different water price policies, there are differences between crops in regard to the cost of water, and changes in the planting area vary, thus affecting the planting structure. Furthermore, regarding the research results of Dong et al. (2020) [2] and Liu et al. (2015) [26], they were based on cross-sectional data to study the impact of different agricultural water price policies on farmers' planting structure or agricultural income. Therefore, the research content and method of this paper are desirable from this point of view.
Second, recently, members of our research team made a return visit to the study area and found that it was not easy to obtain data from farmer household surveys over multiple consecutive periods. In addition to consuming many human and financial resources, the suitability of the study area and the comparability of the research samples should be considered. For example, Linze County adjusted the tiered water price method in 2020, and farmers' operating decisions under the new water price have changed, which makes it difficult to continuously track the impact of a policy over multiple periods. However, if practice and data allow, using dynamic panel data to assess the impact of agricultural water pricing policies holds great significance. Finally, thank you for your comments.
- The basis for variable selection needs further clarification. For example, the selection of control variables should refer to some relevant literature.
Response: Thank you very much for your comment. On the basis of reorganizing and summarizing the relevant literature, we have detailed the selection and classification of variables and supplemented the references [27, 28, 29, 31] on lines 462-480.
- keywords: should be short and should not overlap with the title of the article
- the authors did not present any aim / hypothesis at the beginning
- Conclusion? - the conclusion should not be a summary of discussion. Make sure the conclusion is short and solid. An idea may be to synthetize in 3-5 bullet the key results of the study, evidences and recommendation. This improvement will increase clearness and readability. Add a practical implications statement.
- References Double check name spelling, year, page, etc. I will not check them; it is your responsibility (according to Instructions for Authors)
Reviewer 3 Report
I can confirm that the subject matter of this paper [The effect of the comprehensive reform of agricultural water prices on farmers’ planting structure in the oasis-desert transition zone—A case study of the Heihe River Basin] is of interest and relevance for publication in International Journal of Environmental Research and Public Health
Dear Authors, I read with your interesting manuscript. Here are some revisions to improve it.
- keywords: should be short and should not overlap with the title of the article
- the authors did not present any aim / hypothesis at the beginning
- Conclusion? - the conclusion should not be a summary of discussion. Make sure the conclusion is short and solid. An idea may be to synthetize in 3-5 bullet the key results of the study, evidences and recommendation. This improvement will increase clearness and readability. Add a practical implications statement.
- References Double check name spelling, year, page, etc. I will not check them; it is your responsibility (according to Instructions for Authors)
Author Response
- Keywords: should be short and should not overlap with the title of the article
Response: Based on the reviewer’s comment, we have changed the keywords “Comprehensive reform of agricultural water prices; Tiered water price policy; Farmers’ planting structure; Oasis-desert transition zone in the Heihe River Basin” to "Tiered water price policy; Uniform water price; Agricultural water pricing method; Farmers’ planting structure; Heihe River Basin". Thank you.
- The authors did not present any aim/ hypothesis at the beginning
Response: Thank you very much for your comment. We have carefully considered the content and objectives of this study and introduced the main aim in section 1. Specifically, the aim of this study is to optimize and improve the comprehensive reform of agricultural water prices for water resource management departments, especially for arid and semiarid regions where irrigation agriculture is the mainstay. We discuss the mechanism of the impact of different water price policies and agricultural water pricing methods on the crop planting structure (lines 176-183).
- Conclusion? - the conclusion should not be a summary of discussion.
Make sure the conclusion is short and solid. An idea may be to synthetize in 3-5 bullet the key results of the study, evidences and recommendation. This improvement will increase clearness and readability. Add a practical implications statement.
Response: Thank you for these comments. We have revised the content of the conclusion (lines 656-669). The details are as follows:
Based on farmer survey data on different water price policy implementation areas in the oasis-desert transition zone of the HRB in 2019, this study constructs an analytical framework for the impact of water prices on farmers’ planting structure and builds an econometric model. The crucial findings are as follows. Compared with the uniform water price policy, the implementation of the tiered water price policy will significantly reduce the proportion of high-water-consuming crops planted when other conditions remain unchanged. Additionally, the pricing method of the tiered water price policy will reduce the proportion of farmers planting high-water-consuming crops. However, the difference is not significant, which reveals that when the opportunity cost of irrigation water increases, farmers will increase the proportion of low-water-consuming crops. Finally, the results suggest that the proportion of low-water-consuming crops can be increased by increasing farmers’ educational level, reducing the family-cultivated land area, improving land inflow, increasing the number of crop types, and increasing farmers’ satisfaction with the current subsidy policy.
- References Double check name spelling, year, page, etc. I will not check them; it is your responsibility (according to Instructions for Authors)
Response: Thank you for this comment. We have revised and carefully checked the format of the references based on the instructions for authors.